# Enhanced Performance of Polymer Electrolyte Membranes via Modification with Ionic Liquids for Fuel Cell Applications

**DOI:** 10.3390/membranes11060395

**Published:** 2021-05-27

**Authors:** Jonathan Teik Ean Goh, Ainul Rasyidah Abdul Rahim, Mohd Shahbudin Masdar, Loh Kee Shyuan

**Affiliations:** 1Fuel Cell Institute, Universiti Kebangsaan Malaysia, UKM, Bangi 43600, Selangor, Malaysia; jgoh@ukm.edu.my (J.T.E.G.); ksloh@ukm.edu.my (L.K.S.); 2Department of Chemical and Process Engineering, Universiti Kebangsaan Malaysia, UKM, Bangi 43600, Selangor, Malaysia; zinniagenus@gmail.com

**Keywords:** polymer electrolyte membrane, fuel cell, ionic liquid, proton conductivity, thermal stability

## Abstract

The polymer electrolyte membrane (PEM) is a key component in the PEM fuel cell (PEMFC) system. This study highlights the latest development of PEM technology by combining Nafion^®^ and ionic liquids, namely 2–Hydroxyethylammonium Formate (2–HEAF) and Propylammonium Nitrate (PAN). Test membranes were prepared using the casting technique. The impact of functional groups in grafting, morphology, thermal stability, ion exchange capacity, water absorption, swelling and proton conductivity for the prepared membranes is discussed. Both hybrid membranes showed higher values in ion exchange capacity, water uptake and swelling rate as compared to the recast pure Nafion^®^ membrane. The results also show that the proton conductivity of Nafion^®^/2–HEAF and Nafion^®^/PAN membranes increased with increasing ionic liquid concentrations. The maximum values of proton conductivity for Nafion^®^/2–HEAF and Nafion^®^/PAN membranes were 2.87 and 4.55 mScm^−1^, respectively, equivalent to 2.2 and 3.5 times that of the pure recast Nafion^®^ membrane.

## 1. Introduction

There is a global shift in demand for environmentally friendly and sustainable energy. This has led to increased development of alternative energy resources with lower emissions, higher efficiencies and with potentially inexhaustible supplies. Hydrogen energy and fuel cell technology has been identified as one pathway towards this goal [1]. Generally, a fuel cell generates electricity via chemical reactions with no byproducts other than water and residual heat. Various types of fuel cells exist, including the polymer electrolyte membrane fuel cell (PEMFC), direct methanol fuel cell (DMFC), solid oxide fuel cell (SOFC), alkaline fuel cell (AFC) etc., which are distinguishable from each other based on materials and fuels used. This study focuses on PEMFC technology. 

PEMFC technology was developed mainly for transportation, stationary and portable fuel cell applications [2]. The main components within the PEMFC are two electrodes comprising the anode and cathode, as well as the electrolyte, which is a solid ionomeric membrane. The membrane has the important role of selectively allowing protons to pass through while being electrically insulating. In this area, various works have shown that composite membranes containing ionic liquids, organic–inorganic nanoparticles and metal oxides may improve proton conductivity, which in turn improves the overall performance of fuel cells via reduced internal resistances [3,4,5,6]. 

Nafion^®^ membranes are commonly used in commercial PEMFC applications. However, there are some constraints when using this membrane; among other things the operation of the PEMFC is limited to temperatures not exceeding 100 °C due to evaporation of water molecules in the membrane which would deteriorate its performance [7]. In this respect, incorporating ionic liquids in the Nafion^®^ membrane can also help overcome this problem besides improving the overall performance of the PEMFC in terms of proton conductivity, thermal stability, voltage output and power density [8].

Other researchers have demonstrated the use of ionic liquids as useful additives to membranes for PEMFCs. A polybenzimidazole containing hydroxyl groups (PBIOH)/ionic-liquid-functional silica (ILS) composite membranes (PBIOH–ILS) have been successfully prepared for the high temperature PEMFCs by Wang et al. [9]. The PBIOH-ILS composite membranes have shown excellent thermal stability, oxidative stability and improved mechanical strength. Meanwhile, Nafion/sulfonated poly(ionic liquid) block copolymer (SPILBCP) ionomers were developed by Neyerlin et al. [10]. It resulted in a substantial improvement in MEA performance across the kinetic and mass transport-limited regions. Lin et al. [11] prepared a new type of proton-conducting hybrid membrane that contained protic ionic liquids (PILs) and silica nanoparticles or mesoporous silica nanospheres. The results under anhydrous conditions showed that proton conductivity was up to 1 × 10^−2^ Scm^−1^ at 160 °C; proving that PIL-based hybrid membranes are suitable for high-temperature PEMFC operation. Meanwhile, the synthesis and characterization of proton-conducting ionic liquids (PCILs) and polymer electrolyte membranes (Nafion^®^) was studied by Martinez et al. [12]. Membrane characterization highlighted the advantage of PCIL, where the conductivity of the blended membrane significantly improved even with the addition of a small amount of PCIL. The improvement in proton transport is represented by the cationic transference number, which should be close to 1 since no anionic transfer occurs in the membrane thus the transference number is limited only by the efficiency of proton transport.

Ionic liquids are organic salts with melting points below or equal to room temperature and have special features that are ideal for innovation in membrane fabrication. These include high proton conductivity without the presence of water, low evaporation pressure and high thermal and electrochemical stability, which can reinforce the internal membrane structure [8]. Additionally, ionic liquids are formed entirely of ions, and can be combined to meet the desired characteristics for specific applications [13]. Ionic liquids may be classified as aprotic or protic. Aprotic ionic liquids have high ion mobility and concentration, suitable for lithium battery applications, whereas protic ionic liquids have mobile protons located in cations and are suitable for use as electrolyte in fuel cell applications [14]. A subclass of acidic ILs is the protic ILs, which possess the ability to donate protons or hydrogen ions (H^+^). It has been shown that the incorporation of suitable ionic liquids affected the structure and enhanced the conductivity. For instance, Sood et al. [15] investigated the effect of TEA-TF content on the electrochemical and transport properties of Nafion– triethylammonium trifate (TEA–TF) membranes. The increase in TEA–TF content raised the conductivity and water uptake without affecting the permeability coefficients of oxygen and hydrogen gases. Ding et al. [16] investigated the PIL/functionalized graphene oxide hybrid membranes for high temperature PEMFC applications. It was obtained that incorporating the proper amount of [APMIm][Br]-GO significantly increased the proton conductivity of the hybrid membranes, and the membranes with 1.0 wt.% [APMIm][Br]–GO showed the highest conductivity. With promise in improving the electrolyte properties, therefore, protic ionic liquids are the focus in this study of the PEMFC system.

This work reports the fabrication of polymer electrolyte membranes based on ionic liquid for PEMFC application. The main objective is to improve the internal structure of the polymer electrolyte membrane and produce a membrane that has a robust structure and can withstand high temperature operation, while improving the performance of single cells. In this study, we focused on the different type of protic ILs such as 2–HEAF and PAN incorporated with Nafion in order to investigate other possible PILs and the effect of these ILs for electrolyte membrane in PEMFC. The polymer electrolyte membrane is formulated using Nafion^®^ and ionic liquids, which are 2–HEAF and PAN. All membranes in this study were prepared using the casting method. 

Membrane characterization was performed using scanning electron microscopy (SEM), and Fourier-transform infrared spectroscopy (FTIR). The membrane thermal stability and conductivity was evaluated using thermogravimetric analysis (TGA), ion exchange capacity (IEC), water uptake and swelling and electrochemical impedance spectroscopy (EIS). The membrane electrode assembly (MEA) was fabricated to evaluate the single cell performance in the PEMFC application. Similar characterization and performance analysis tests were carried out on the produced membrane, Nafion^®^/2–HEAF and Nafion^®^/PAN, as well as on recast Nafion^®^ membrane for reference. Based on the experimental results, the effects of ionic liquids on the polymer electrolyte membrane for PEMFC application are discussed.

## 2. Materials and Methods

### 2.1. Materials

The ionic liquids of 2–hydroxyethylammonium formate (2–HEAF) and propylammonium nitrate (PAN) were purchased from IoLiTec Ionic Liquids Technologies GmbH, with >97% purity. Nafion^®^ 112 (5%), dimethylformamide (DMF (98%)) and platinum (Pt) black catalyst, were obtained from Sigma-Aldrich Co., Ltd. All the materials were used as received without further purification. The chemical formulas for Nafion^®^ 112, 2–HEAF and PAN are C_7_HF_13_O_5_S.C_2_F_4_, C_3_H_9_NO_3_ and C_3_H_10_N_2_O_3_, respectively, and the chemical structures are shown in Figure 1. 

### 2.2. Preparation of Polymer Electrolyte Membranes (PEM)

The fabrication of the Nafion^®^/ionic liquid membrane involved several steps that included drying, mixing, and casting. First, 112 Nafion^®^ solution was dried in a fume chamber for 24 h to evaporate the alcohol that is pre-mixed with the Nafion^®^ solution. Then, the solid Nafion^®^ was dissolved using DMF solvent, which accounted for 88% of the liquid content in Nafion^®^ 112. The ionic liquid was later mixed into the Nafion^®^ solution and stirred for 3 h at 50 °C, until the mixture became homogenous. The mixing of ionic liquid and Nafion^®^ solution involved certain percentage ratios, with 40 and 70 wt.% of ionic liquid in 1 g of Nafion^®^ solution used in this study. The mixture was casted into a petri dish for slow solidification without any formation of foam. The solution was solidified at a temperature range of 50–80 °C for 4 h in oven. The produced membrane was peeled off from the petri dish and ready to be used for performance testing.

### 2.3. Characterization Methods

#### 2.3.1. Morphology, Functional Groups and Thermal Properties

Surface morphology of the prepared membrane was viewed using SEM (Zeiss/SUPRA 55 VP, Oberkochen, Germany). FT–IR analysis (Thermo Scientific, Waltham, MA, USA) provided information regarding the presence of functional groups in evaluating the grafting process between Nafion^®^ polymer and ionic liquid. The thermal stability of the membrane was tested via TGA (Shimadzu TGA 50, Kyoto, Japan). In addition, TGA was also used to provide information on the change in weight loss patterns as a function of temperature which alluded to the robustness of the membrane. These analyses were applied to all samples, recast Nafion^®^ and Nafion^®^/ionic liquid membranes.

#### 2.3.2. IEC, Water Uptake and Swelling

IEC values were determined by a back-titration method. The membrane in acid form was immersed in 0.3 M NaCl solution for 24 h in order to exchange sodium ions with the protons. Then, a few drops of phenolphthalein which acted as an indicator were added to the solution with the membrane and followed by the titration method using 0.1 M NaOH solution. The IEC values were calculated by Equation (1):IEC (mmol/g) = ((V_NaOH final_ − V_NaOH initial_) × (M_NaOH_))/W_m_(1)
where V_NaOH final_ and V_NaOH initial_ are the final and initial volume (mL) of the NaOH solution, respectively, M_NaOH_ is the molar concentration (M) of the NaOH solution and W_m_ is the weight (g) of the dried membrane.

The water uptake measurement was evaluated by the weight and dimensional change of the membrane from wet to dry state. The membrane samples were dried in the oven until the weight was constant. Then, the membrane was immersed in deionized water for 24 h at room temperature to achieve equilibrated water uptake. The water uptake of the polymer based ionic liquid membrane was calculated using the Equation (2), where W_wet_ and W_dry_ are the weights of the wet and dry membranes:WU = ((W_wet_ − W_dry_)/W_dry_) × 100(2)

Swelling measurements were also recorded simultaneously and were calculated according to Equation (3). Where, t_wet_ and t_dry_ are the length, thickness or volume of wet and dry membranes, while S is the swelling volume of the immobilized ionic liquid membranes immersed in water for 24 h.
S = ((t_wet_ − t_dry_)/t_dry_) × 100(3)

#### 2.3.3. Proton Conductivity

The proton conductivity of the immobilized ionic liquid membranes was measured via EIS testing using a potentiostat/galvanostat (VersaSTAT 4). The supported membranes were fixed in conductivity cells that comprised two outer platinum foils and two inner platinum wires. The potentiostat/galvanostat was set to apply a specific voltage between the two inner probes and measure the resulting current. The slope of Current vs. Voltage plot was used to determine the resistance, R (Ω) and proton conductivity (σ). The conductivity values were derived using Equation (4), where d (cm) and A (cm^2^) are the thickness and surface area of the sample:σ (Scm^−1^) = d/(R × A)(4)

#### 2.3.4. Single Cell Performance

The MEA was prepared by placing the Nafion^®^ and Nafion^®^/ionic liquid membranes directly between two electrodes for supply of hydrogen gas at the anode and air on the cathode. The electrodes consist of two functional layers i.e., Toray carbon paper as the gas diffusion layer (GDL) and a catalyst layer. The two-layers were prepared using casting method. The GDL layer consisted of 0.1 g carbon black, 2 g 2–propanol and 0.22 g Nafion^®^ solution. Materials were mixed and homogenized in ultrasonic cell crusher for 15–20 min. The casting method was used and the GDL layer was heated at a temperature of 100 °C for 1 h. 

For the catalyst layer, the catalyst loading used was 0.25 mg_Pt_cm^−2^ and 5 mg_Pt_cm^−2^ for the anode and cathode, respectively. The active area used was calculated to be 4.84 cm^2^. This layer was fabricated on top of the GDL layer. The catalyst, Nafion^®^ solution (10 wt.% of catalyst used), 1100 mL of 2–propanol and 300 mL of deionized water were mixed and homogenized. The casting method was carried out using special brush to obtain a flat and smooth surface. The second layer was heated in the oven at 90 °C for 1 h. The MEA components were affixed by hot press machine with 14 bar pressure and 135 °C temperature for 3 min. Finally, the MEA was incorporated into with a single cell PEMFC assembly and tests were carried out in a controlled temperature room at 25 °C (ambient), hydrogen gas flow rate of 300 mLmin^−1^ and 3000 mLmin^−1^ for air. The performance of the single cell PEMFC with different MEAs was tested and obtained using an electrochemical measurement device, Fuel Cell Monitor Pro 3.0. 

## 3. Results and Discussion

### 3.1. Morphology of Membranes

Surface morphology and thickness of the membranes were obtained from SEM analysis. Figure 2 shows the SEM images for all membrane samples at a magnification of 5 kX. Based on these images, it is noted that the recast Nafion^®^ membrane has a smoother and uniform surface as compared to the other membranes.

Based on the images obtained, the surface morphology of both ionic liquid-based membranes displayed patches of white crystal-like structures due to the presence of mixed ionic liquids [17]. Figure 2b shows that the 2–HEAF ionic liquid is dispersed quite uniformly within the Nafion^®^ membrane at 40 wt.%. The high degree of uniformity may help in reducing the occurrence and extent of variations on the membrane that could often cause the local hot spots that lead to the holes in the membrane [18]. Meanwhile, minor agglomeration was observed on the Nafion^®^/2–HEAF (70 wt.%) membrane with both Nafion^®^/2–HEAF membranes exhibiting some roughness on the membrane surface. The Nafion^®^/PAN membrane (40 wt.%) presented the most well-dispersed ionic liquid in the Nafion^®^ membrane, with several small void spaces and pinholes. However, the 70 wt.% of Nafion^®^/PAN membrane showed the opposite result, with visible wrinkling through the cross section of the membrane due to the high loading of the ionic liquid [19].

Table 1 shows the resulting thickness of membranes obtained from the SEM analysis for the recast Nafion^®^ and produced Nafion^®^/ionic liquid membranes. The recast Nafion^®^ membrane had the thinnest membrane with 62.53 μm, followed by Nafion^®^/2-HEAF (40 wt.%) and Nafion^®^/PAN (40 wt.%) membrane with 66.63 and 112.07 μm. The thickness of the membrane is one of the main parameters that affect the PEMFC performance. Generally, PEMFCs with thinner membranes show better performance than those with thicker membranes [20,21]. This is because the membrane thickness may change ohmic and concentration loss in the I–V curves [20]. The thinner membrane also has less internal resistance and facilitates the proton transfer through the membrane more quickly, which leads to better fuel cell performance [20,21]. Thus, the recast Nafion^®^ and Nafion^®^/2-HEAF (40 wt.%) are expected to have better performance in single cell test.

### 3.2. FT–IR Analysis

FT–IR analysis was conducted to identify the existence of functional groups in the recast Nafion^®^ and Nafion^®^/ionic liquid membrane. The IR spectrum for recast Nafion^®^, Nafion^®^/2-HEAF (40 and 70 wt.%) and Nafion^®^/PAN (40 and 70 wt.%) membrane is presented in Figure 3. Overall, the IR spectrum results illustrate that the Nafion^®^/ionic liquid have almost the same peaks compared to the recast Nafion^®^ membrane. Some transition and new peaks were also observed on Nafion^®^/ionic liquid spectrum due to the addition of ionic liquid composition in the membrane.

The IR spectrum for all membranes shows the existence of O–H functional group, where the peak for recast Nafion^®^, 40 and 70 wt.% of Nafion^®^/2–HEAF and 40 and 70 wt.% of Nafion^®^/PAN membrane appeared at 3096, 3114, 3112, 3112 and 3091 cm^−1^, respectively. This hydroxyl functional group directly facilitates the permeation of H^+^ ions for proton conductivity through the membrane. Figure 3 also shows some new peaks for the Nafion^®^/ionic liquid membranes which are not present in the recast Nafion^®^ membrane. For all Nafion^®^/ionic liquid membrane samples, the spectrum for C=N lies between the wave numbers 1613–1624 cm^−1^, while regions of lower intensity located between 1513 and 1521 cm^−1^ indicated the presence of N–H functional groups corresponding to the presence of ammonium base found in ionic liquids.

Strong and sharp regions at wave numbers 1227–1315 cm^−1^ correspond to the C–N functional group. Closer observation shows that the C–N peak for the recast Nafion^®^ membrane is at 1227 cm^−1^, while for Nafion^®^/ionic liquid membrane is in the range of 1228–1315 cm^−1^. This transition peak occurs due to the mixing process and ratio of the ionic liquid to the Nafion^®^. The presence of the C–C–C=O functional group was also identified with lesser intensity within the wave number 1046–1156 cm^−1^.

At the wavelength in between 500–1000 cm^−1^, two functional groups appeared that consisted of C–H (961–982 cm^−1^) and C–Br (511–633 cm^−1^). Both functional groups also showed a transition peak between the recast Nafion^®^ and the Nafion^®^/ionic liquid membranes, caused by the ratio of Nafion^®^ and ionic liquids. The presence of C–H bonds suggests interactions between the Nafion^®^, ionic liquids and DMF solvents, meanwhile C-Br bonds occur due to mixed reactive reactions in the formation of polymer electrolyte membrane which increases with temperature.

In general, both Nafion^®^/ionic liquid membranes have the same functional groups and show similar modifications in terms of transition and additional peaks compared to the recast Nafion^®^ membrane. The bonds of O–H, C–N, C–C–C=O, C–H, and C–Br are formed during membrane production because of the reaction between Nafion^®^, ionic liquids and DMF solvent. The presence of ionic liquids in the membranes are identified by the additional peak of N–H and C=N at 1513–1624 cm^−1^. The addition of ionic liquids causes changes in the original Nafion^®^ membrane structure and alkyl groups contained in the ionic liquid form hydrogen bonds with sulfonic acid groups in Nafion^®^ cation. The C–C–C=O and C=N bonds further reinforce internal structure of the membrane [22].

### 3.3. TGA Analysis 

The membrane samples underwent a thermal stability test as a primary qualifier for membrane durability in PEMFC operation. The graph of the weight loss (%) against temperature (°C) for all the membranes is given in Figure 4. The thermal stability of the membranes was analyzed as a function of the temperature, which was in the range of 25–600 °C.

Based on the TGA thermograms all samples were stable up to 200 °C. At the range of 200–230 °C, the recast Nafion^®^ membrane shows the sharp drop pattern on the weight loss, while the other Nafion^®^/ionic liquid remained stable. This indication of the first stage of weight loss, below 250 °C is attributed primarily to the evaporation of water molecules [23]. This proves that the incorporation of ionic liquids can improve the membrane thermal resistance to temperature variations. This advantage can be traced to the formation of the ionic pair –SO_3_^−^NH_4_^+^, one of the active Nafion^®^ ions in ionic liquids, which stabilizes the bonds in the membrane [24]. 

Up to 350 °C, the Nafion^®^/ionic liquid membranes were observed to retain their weight at about 95%, while the recast Nafion^®^ membrane weight dropped to about 80%. However, all the membranes begin to lose their weight significantly beyond temperature likely due to the loosening of sulfonic acid groups present in the Nafion^®^ membrane [25]. It is notable that the onset temperature of membrane weight loss varies from one another. The sharp weight loss for Nafion^®^/2-HEAF membrane occurred in the range of 330–340 °C, followed by Nafion^®^/PAN (347–350 °C) and recast Nafion^®^ membrane (360 °C) until the temperature of about 400 °C. At this point, the weight loss for recast Nafion^®^ membrane was 59.31%, while the Nafion^®^/ionic liquid was 73.73–68.8% on average. This is because the anti-sulfonation active ionic bonds that occur between the Nafion^®^ molecule on the membrane at active site Bronsted acid and base along with sulfonation on Nafion^®^ bond began to unravel [24]. After the sudden drop, all the membranes show a slight decline before stabilizing at 400–450 °C. This upgrade may be due to the interaction between the polymer sulfonic acid and hydroxyl group, which enhances the membrane heat resistance and reduces the rate of decomposition [23].

The next stage involved increasing the temperature beyond 450 °C, correlating to the degradation of the polymer backbone [23,24]. At this stage, all the membranes showed severe weight loss and stabilized afterwards at 500 °C. At 600 °C, all the membrane had decomposed, and the remaining mass was different across all samples. The order of membrane residual mass is as follows: Nafion^®^/2–HEAF (40 wt.%) > recast Nafion^®^ > Nafion^®^/PAN (70 wt.%) > Nafion^®^/2–HEAF (70 wt.%) > Nafion^®^/PAN (40 wt.%). This order shows that the Nafion^®^/2–HEAF (40 wt.%) membrane has the highest thermal stability as compared the other membranes, which proved that the ionic liquids improved the thermal behavior of the membrane.

### 3.4. IEC, Water Uptake and Swelling 

The IEC analysis was carried out to identify the concentration of free charge carriers in the membrane, meanwhile the water uptake and swelling analyses were to test the capabilities of membrane in water [23,26]. The results of these analyses for the recast Nafion^®^, Nafion^®^/2–HEAF and Nafion^®^/PAN membrane are tabulated in Table 2. The results are shown as mean values, as all measurements were repeated until consistent results were obtained. 

The results show that the Nafion^®^/PAN (40 wt.%) has the highest IEC value of 2.4 mmolg^−1^. The other ionic liquid-based membranes also exhibit high IEC value with 2.1, 1.9, and 1.6 mmolg^−1^ for Nafion^®^/PAN (70 wt.%), Nafion^®^/2–HEAF (40 wt.%), Nafion^®^/2–HEAF (70 wt.%), respectively. The recast Nafion^®^ membrane has the lowest IEC value of 1.4 mmolg^−1^. Based on the results obtained, the ionic liquid-based membranes have a higher IEC value than recast Nafion^®^ membrane, due to the presence and effectiveness of ionic liquids incorporated with Nafion^®^ polymer that increased the concentration of free charge carriers in the membrane.

This can be explained by the strength of the bond between functional groups of polymer Nafion^®^ and the ionic liquid due to the presence of alkyl group as found in FT–IR analysis, which produced stronger structural bonds with polymer Nafion^®^ [22]. The chemical structure of ionic liquids is influenced by the degree of ionization protic. Walden Regulation Classic concept can be used to explain this phenomena: the incorporation of ionic liquids with the polymer Nafion^®^ positively impacts the degree of ionization in its internal structure and thus increases proton conductivity [27]. However, based on the overall result, the ionic liquid loading appears to be beneficial up to a certain percentage only beyond which excessive loading may reduce the IEC value. Therefore, further study is required to determine the optimum loading of ionic liquids that can incorporate with Nafion^®^ polymer.

The membrane response towards water was evaluated using the water uptake and swelling analysis. The results for water uptake and swelling are in terms of percentage (%) and membrane thickness, S (cm). The results of the analysis are tabulated in Table 2 and it was found that the increased ratio of incorporated ionic liquid improved the membrane ability to absorb water and accommodate further membrane thickness. With only 1.1% difference, the highest water uptake was demonstrated by Nafion^®^/2–HEAF and Nafion^®^/PAN membrane at 70 wt.% of ionic liquid in the range of 36.3–37.2%. The Nafion^®^/ionic liquid membrane at this percentage also showed significant change in membrane size expansion compared with other membranes, where the measurement S_x_, S_y_ and overall volume are the highest. The other Nafion^®^/ionic liquid membranes show the higher percentage of water uptake and membrane size expansion as compared to the recast Nafion^®^ membrane.

The capability of the membrane for water absorption is related to membrane size expansion and closely linked to the ability of a membrane to form strong internal bonds, thus the degree of internal bond formation on the Nafion^®^/ionic liquid membrane structure is higher than the recast Nafion^®^ membrane. The membrane hydrophobic surface also affects water absorption and membrane expansion. The increased percentage of ionic liquid reduced hydrophobicity of the membrane surface. The water absorption phenomena on the membrane surface is the first step in creating favorable membrane conditions that improve the conductivity protons and ion exchange capacity [28].

### 3.5. Proton Conductivity

Proton conductivity is the primary function of the membrane and is evaluated using EIS analysis. All the membranes were analyzed in their dehydrated and hydrated state with a 1.76 cm^2^ electrode surface area. Table 3 shows the overall results of proton conductivity in dehydrated and hydrated states for recast Nafion^®^, Nafion^®^/2–HEAF (40 and 70 wt.%) and Nafion^®^/PAN (40 and 70 wt.%) membrane.

The EIS results for dehydrated state show that the Nafion^®^/PAN (40 wt.%) membrane produced the highest proton conductivity with a value of 5.05 × 10^−3^ Scm^−1^ and followed by Nafion^®^/PAN (70 wt.%), Nafion^®^/2–HEAF (40 wt.%), recast Nafion^®^ and Nafion^®^/2–HEAF (70 wt.%) membrane with a value between 5.05 × 10^−3^ and 9.54 × 10^−4^ Scm^−1^. This alludes to the bonding of the ammonium base group between the carbon bonds in the membrane being more robust at 40 wt.% for Nafion^®^/PAN membrane than the other membranes. The overall trend on dehydrated state results indicates that the percentage of ionic liquid used in the membrane affects the value of proton conductivity. When considering both the Nafion^®^/ionic liquid membranes, the proton conductivity decreases at the percentage of 70 wt.% ionic liquid. This is due to the bonding disorder between the functional groups found in the Nafion^®^ and ionic liquids, resulting from active bonding sites in each functional group reaching saturation.

For the hydrated state, the membrane proton conductivity in descending order is as follows: Nafion^®^/PAN (40 wt.%) > Nafion^®^/PAN (70 wt.%) > Nafion^®^/2–HEAF (40 wt.%) > Nafion^®^/2–HEAF (70 wt.%) > recast Nafion^®^ membrane. Proton conductivity of Nafion^®^/PAN (40 wt.%) membrane is approximately four times greater than the recast Nafion^®^ membrane. The EIS results also show that the proton conductivity in the hydrated state is higher and more stable than the dehydrated state. The water properties as a conductor is one of the main factors that contribute to this improvement. The presence of ionic liquids in the Nafion^®^ can increase the absorption capacity of water and the presence of water has a positive impact towards proton conductivity. This is proven by the proton conductivity result of the Nafion^®^/ionic liquid membrane that is in line with the results obtained in the IEC, water uptake and swelling analysis.

Overall, all four ionic liquid membranes have higher conductivity than recast Nafion^®^ membrane. These results correspond to a number of recent studies that found that the combinations of Nafion^®^ and ionic liquids, namely Nafion^®^/[EMIM]Cl and Nafion^®^/PBI/DESH demonstrated higher proton conductivity compared to the pure Nafion^®^ membrane [29,30]. This indicates that the introduction of ionic fluids in the membrane has a positive effect on the proton conductivity. The increase in conductivity is due to the structure in the protic elements of the ionic liquid that joined with the Nafion^®^. Nafion^®^ provides a proton pathway within its structure and the sulfonic acid ions combine with the acid ions present in ionic liquids through the hopping mechanism. The random motion in the ionic absorption between the bonds will be more effective, which is at optimum levels for the membrane [31,32].

### 3.6. MEA Performance

The MEA performance including the polarization curve (a) and power curve (b) using the different membranes is shown in Figure 5. A single cell PEMFC with 4.84 cm^2^ active area was used at ambient condition. Based on the Figure 5a, the open circuit voltages (OCVs) for all membranes are higher than 0.9 V and the cell voltage decreases with an increase in the current applied, suggesting that all the membranes exhibit positive tolerance to the fuel gas of hydrogen and oxygen. 

The polarization profiles are expected to be similar to the previous works for PEMFC and showed good agreement in terms of activation, ohmic and mass transport losses. For the power profiles as shown in Figure 5b, the MEA with Nafion^®^/PAN (40 wt.%) membrane shows the best maximum power density which was only 4.2% lower than the commercial Nafion^®^ 112. This performance validates the use of PAN ionic liquid blended with Nafion^®^ membranes and has a potential to be used for PEMFC due to higher stability as shown in TGA analysis. The order of peak power densities is shown in Table 4 below.

The MEA performance showed good correlation with the proton conductivity of the membranes shown in Table 3 and follows the same trend. The interface between electrolyte and electrocatalyst plays an important role in determining the activation polarization loss; the greater the interfacial contact between the electrocatalyst and the electrolyte, the lesser activation losses suffered [33]. The electrolyte ‘quality’ in this case is determined by the active sites and the proton cross linkage of the membranes which is highest for MEA with Nafion^®^/PAN (40 wt.%) membrane. In terms of ionic liquid content, 40 wt.% loading is well dispersed throughout the membrane compared to that 70 wt.%, leading to more uniform and homogeneous distribution of nanoparticles. The homogeneous distribution leads to better adsorption of the electrolyte leading to good electrolyte electrocatalyst contact and thus lesser activation loss [33]. The increased mass transport for ionic liquid of PAN and 40 wt.% loads can be explained by the increased hopping sites for proton transport which will contribute towards higher current density during MEA testing.

## 4. Conclusions

The 2–HEAF and PAN ionic liquids were successfully incorporated into the Nafion^®^ membrane using the casting technique. Both the ionic liquid-based membranes were compared with a recast Nafion^®^ membrane, which is typically used for commercial PEMFC devices. The type and amount of ionic liquids, for example 40 and 70 wt.%, affect the properties of the membrane. The 2–HEAF membranes showed some surface roughness while a higher wt.% of ionic liquid exhibited some agglomeration of particles. The IEC and proton conductivity of the Nafion^®^/PAN (40 wt.%) membrane was the highest at 8.55 mScm^−1^, which was about four times higher than the recast Nafion^®^ membrane. In terms of power density, the MEA with Nafion^®^/PAN (40 wt.%) membrane shows the highest performance (94.14 mWcm^−2^) compared to the other synthesized membranes which was only 4.2% lower as compared to the commercial Nafion^®^ 112. In the case of ionic liquids, 40 wt.% loading showed better dispersion throughout the membrane compared to that 70 wt.%, leading to a more uniform and homogeneous distribution of the nanoparticles. Uniform and homogeneous distributions lead to better adsorption of the electrolyte which improved electrolyte-to-electrocatalyst contact and thus reduced activation losses as evidenced by the higher power density using 40 wt.% of ionic liquid. In summary, the introduction of ionic liquids in the polymer electrolyte membrane has significant potential for PEMFC application; even as further optimization is still required.

## Figures and Tables

**Figure 1 membranes-11-00395-f001:**
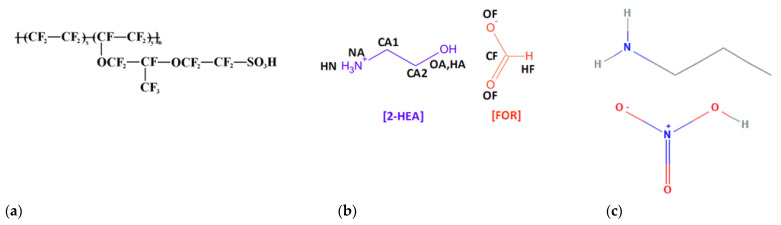
Chemical structure of (**a**) Nafion^®^ 112, (**b**) 2-HEAF and (**c**) PAN.

**Figure 2 membranes-11-00395-f002:**
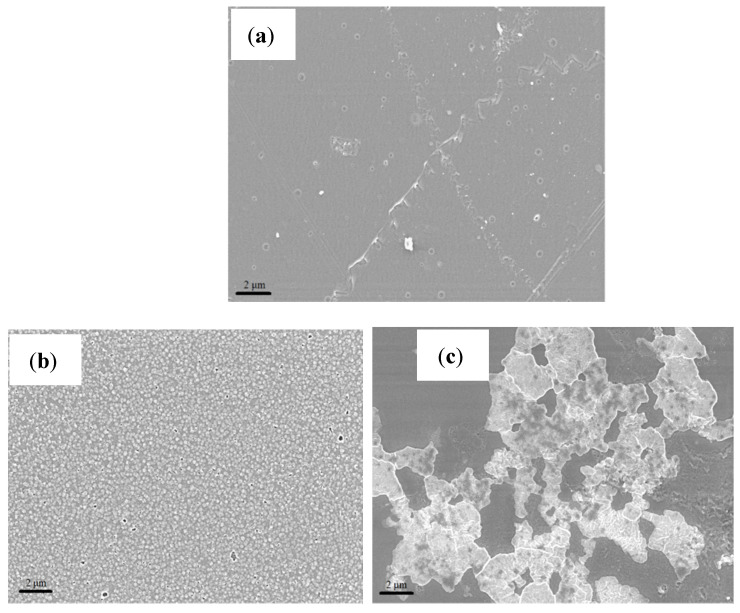
SEM image of the PEM at magnification of 5 kX (**a**) recast Nafion^®^, (**b**) Nafion^®^/2-HEAF (40 wt.%), (**c**) Nafion^®^/2-HEAF (70 wt.%), (**d**) Nafion^®^/PAN (40 wt.%) and (**e**) Nafion^®^/PAN (70 wt.%).

**Figure 3 membranes-11-00395-f003:**
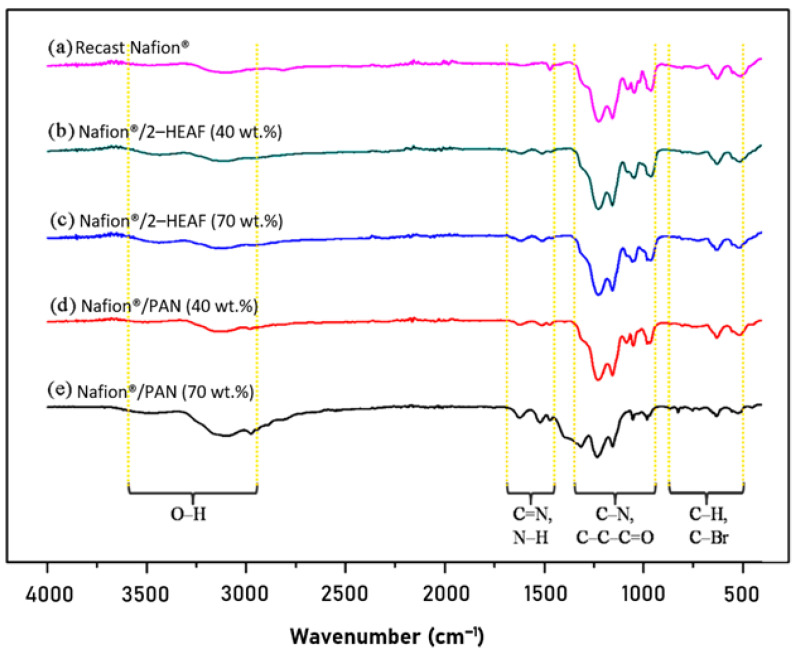
IR spectrum of membranes (**a**) recast Nafion^®^, (**b**) Nafion^®^/2–HEAF (40 wt.%), (**c**) Nafion^®^/2–HEAF (70 wt.%), (**d**) Nafion^®^/PAN (40 wt.%) and (**e**) Nafion^®^/PAN (70 wt.%).

**Figure 4 membranes-11-00395-f004:**
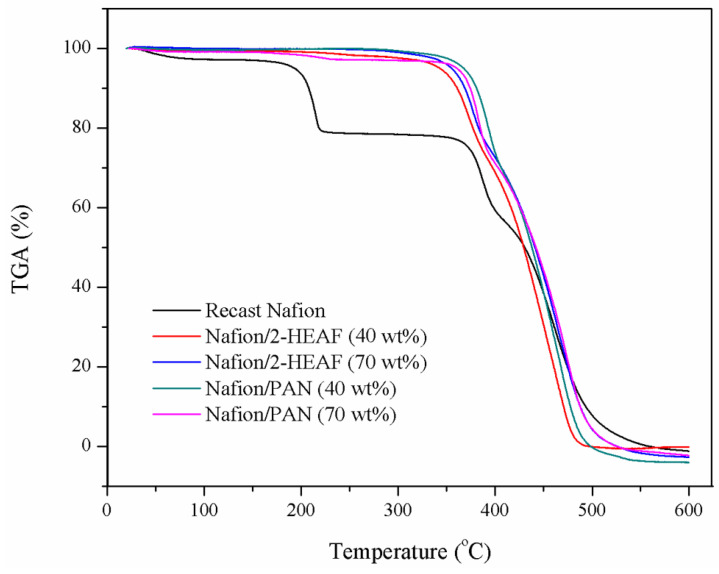
Thermal stability tests for all the membranes.

**Figure 5 membranes-11-00395-f005:**
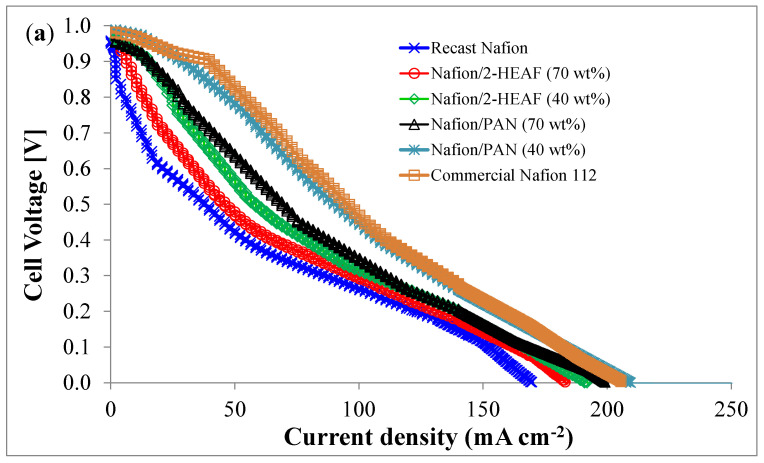
Single cell PEMFCs performance at different types of MEA using various synthesized membranes: (**a**) polarization curve and (**b**) power curve.

**Table 1 membranes-11-00395-t001:** Thickness of membrane from cross-sectional SEM analysis.

Type of Membrane	Thickness (μm)
Recast Nafion^®^	62.53
Nafion^®^/2–HEAF (40 wt.%)	66.63
Nafion^®^/2–HEAF (70 wt.%)	115.57
Nafion^®^/PAN (40 wt.%)	112.07
Nafion^®^/PAN (70 wt.%)	300.37

**Table 2 membranes-11-00395-t002:** Summarized result of IEC, water uptake and swelling analysis for all membrane.

Membrane	IEC (mmolg^−1^)	Water Uptake (%)	Swelling (cm)			
			*S_x_*	*S_y_*	*S_z_*	Volume (m^3^)
Recast Nafion^®^	1.4	24.2	1.05	0.01	0.011	0.0022
Nafion^®^/2–HEAF (40 wt.%)	1.9	28.5	1.30	0.02	0.027	0.0091
Nafion^®^/2–HEAF (70 wt.%)	1.6	37.2	1.37	0.02	0.031	0.0151
Nafion^®^/PAN (40 wt.%)	2.4	27.4	1.22	0.02	0.025	0.0061
Nafion^®^/PAN (70 wt.%)	2.1	36.3	1.33	0.02	0.030	0.0133

**Table 3 membranes-11-00395-t003:** Proton conductivity and resistance for all membranes in dehydrated and hydrated states.

Membrane	Dehydrated State		Hydrated State	
	Resistance (ohm)	Conductivity (S cm^−1^)	Resistance (ohm)	Conductivity (S cm^−1^)
Recast Nafion^®^	4.60	1.23 × 10^−3^	4.36	2.10 × 10^−3^
Nafion^®^/2–HEAF (40 wt.%)	3.86	2.87 × 10^−3^	3.12	5.81 × 10^−3^
Nafion^®^/2–HEAF (70 wt.%)	4.90	0.95 × 10^−3^	3.75	3.03 × 10^−3^
Nafion^®^/PAN (40 wt.%)	2.40	5.05 × 10^−3^	1.87	8.55 × 10^−3^
Nafion^®^/PAN (70 wt.%)	3.37	3.05 × 10^−3^	2.34	6.60 × 10^−3^

**Table 4 membranes-11-00395-t004:** Peak power density and limiting current density for all membrane types.

Membrane	Peak Power Density (mWcm^−2^)	Limiting Current Density (mAcm^−2^)
Recast Nafion^®^	26.64	169.26
Nafion^®^/2–HEAF (40 wt.%)	31.87	191.91
Nafion^®^/2–HEAF (70 wt.%)	29.89	183.08
Nafion^®^/PAN (40 wt.%)	47.09	209.52
Nafion^®^/PAN (70 wt.%)	38.50	202.47
Commercial Nafion^®^ 112	49.15	205.54

## Data Availability

Not applicable.

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
