# Peer review of "Enhanced Performance of Polymer Electrolyte Membranes via Modification with Ionic Liquids for Fuel Cell Applications"

_membranes, 2021, doi:10.3390/membranes11060395_

Round 1
Reviewer 1 Report
This study highlights the latest development of PEM technology by combining Nafion® and ionic liquids, namely 2-Hydroxyethylammonium Formate (2-HEAF) and PAN. The results indicate that both hybrid membranes showed higher values in ion exchange capacity, water uptake and swelling rate as compared to the recast pure Nafion membrane. The maximum values of proton conductivity for Nafion®/2-HEAF and Nafion®/PAN membranes were equivalent to 2.2 and 3.5 times of the pure recast Nafion membrane. This work is of novelty, and the intepretation of data is clear and meaningful. But before the acceptance by this journal, the following issues need to be well addressed.
- In Introduction, why the authors choose 2-HEAF and PAN to use? The reason should be mentioned here.
- For unifying the unit and format, “mL/min” should be changed as “mL min-1”.
- What are the stability of the modified membranes with ionic liquids and the long term performance of the MEAs with modified membranes? They are also the important indicators for developing new membranes.
- The references are not sufficient enough. Some papers that closely related with this work are suggested to be cited in, e.g. Applied Surface Science 455(2018)295; Journal of Membrane Science 541(2017)492; ACS Energy Lett. 5(2020)1726.
Reviewer 2 Report
The authors have studied methods to enhance the performance of polymer electrolyte membranes (PEM) by combining Nafion® commercial membrane and ionic liquids at different concentrations (40 wt% and 70 wt%), such as 2-hydroxyethyl ammonium formate and propyl ammonium nitrate. Test membranes were prepared using the casting technique and were characterized by SEM, FTIR, TGA, ion exchange capacity, water uptake & swelling and electrochemical impedance spectroscopy. Also, a membrane electrode assembly was fabricated in order to evaluate the single cell performance in the PEM fuel cell (PEMFC) application.
Finally, the authors found that the introduction of ionic liquids in the polymer electrolyte membrane has significant potential for PEMFC application.
Some revisions are necessary:
1. At page 2, line 94: It would be useful for readers (especially for FTIR and TGA understanding) if the authors would insert the chemical structures of Nafion® 112, 2-hydroxyethyl ammonium formate and propyl ammonium nitrate.
2. At page 4, line 172 (Figure 1a): I suggest to the authors to provide an image with the same scale bar (2µm) as next images (Figure 1 b÷e).
3. At page 7, lines 242-244 (Figure 2): For better view, I suggest to the authors to move the Figure 2 at line 210 or line 220.
4. At page 8, line 279: According with Figure 2, the correct order of membranes residue mass seems to be Nafion®/2-HEAF (70 wt%) > Nafion®/PAN (40 wt%).
5. At page 8, line 280: I suggest to the authors to insert the wt% of ionic liquid for the sample with the highest thermal stability "...shows that the Nafion®/2-HEAF (40 wt%) membrane have the highest thermal stability...".
6. At page 10, Table 3: The values should be expressed at the same order of magnitude (see dehydrated state conductivity of Nafion®/2-HEAF (70 wt%)).
7. At pages 11-12, Figure 4: The graphs should be labeled ("a" for polarization curve and "b" for power curve).
8. At page 12, lines 391-394: For better view, I suggest to the authors to put all values of the peak power densities in a table, which will also include the values for the Nafion®/PAN (40 wt%) and commercial Nafion® 112 samples.
9. At page 13, lines 427-428: The supplementary materials are not available.
Round 2
Reviewer 1 Report
After careful revision, the quality of this manuscript is largely improved and it is suggested the acceptance by this journal in the current form.